# Whey Protein Drink Ingestion before Breakfast Suppressed Energy Intake at Breakfast and Lunch, but Not during Dinner, and Was Less Suppressed in Healthy Older than Younger Men

**DOI:** 10.3390/nu12113318

**Published:** 2020-10-29

**Authors:** Avneet Oberoi, Caroline Giezenaar, Alina Clames, Kristine Bøhler, Kylie Lange, Michael Horowitz, Karen L. Jones, Ian Chapman, Stijn Soenen

**Affiliations:** 1Adelaide Medical School and Centre of Research Excellence in Translating Nutritional Science to Good Health, The University of Adelaide, Adelaide, Royal Adelaide Hospital, Adelaide, SA 5000, South-Australia, Australia; avneet.oberoi@adelaide.edu.au (A.O.); alina.clames@adelaide.edu.au (A.C.); kristine.bohler@adelaide.edu.au (K.B.); kylie.lange@adelaide.edu.au (K.L.); michael.horowitz@adelaide.edu.au (M.H.); karen.jones@adelaide.edu.au (K.L.J.); ian.chapman@adelaide.edu.au (I.C.); 2Riddet Institute, Massey University, Palmerston North 9430, New Zealand; c.giezenaar@massey.ac.nz; 3Faculty of Health Sciences & Medicine, Bond University, Gold Coast 4229, Queensland, Australia

**Keywords:** whey protein, energy intake, gastric emptying, appetite

## Abstract

Ageing is associated with changes in feeding behavior. We have reported that there is suppression of energy intake three hours after whey protein drink ingestion in young, but not older, men. This study aimed to determine these effects over a time period of 9 h. Fifteen younger (27 ± 1 years, 25.8 ± 0.7 kg/m^2^) and 15 older (75 ± 2 years, 26.6 ± 0.8 kg/m^2^) healthy men were studied on three occasions on which they received, in a randomized order, a 30 g/120 kcal, 70 g/280 kcal whey-protein, or control (~2 kcal) drink. Ad-libitum energy intake (sum of breakfast, lunch, and dinner) was suppressed in a protein load responsive fashion (*P* = 0.001). Suppression was minimal at breakfast, substantial at lunch (~−16%, *P* = 0.001), no longer present by dinner, and was less in older than younger men (−3 ± 4% vs. −8 ± 4%, *P* = 0.027). Cumulative protein intake was increased in the younger and older men (+20% and +42%, *P* < 0.001). Visual analogue scale ratings of fullness were higher and desire to eat and prospective food consumption were lower after protein vs. control, and these effects were smaller in older vs. younger men (interaction effect *P* < 0.05). These findings support the use of whey-protein drink supplements in older people who aim to increase their protein intake without decreasing their overall energy intake.

## 1. Introduction

The number of older people with malnutrition, both under- and over-nutrition, is rising [1]. Healthy ageing is associated with a reduction in appetite and food intake, including protein intake, which predisposes older people to loss of body weight and in particular, skeletal muscle mass [2,3]. The latter is associated with a decrease in function and quality of life [4]. The causes of the reduction in food intake during healthy ageing are likely to be heterogeneous, including changes in gastrointestinal mechanisms induced by nutrient intake, such as slowing of gastric emptying [5,6].

A common strategy to increase energy intake and body weight in undernourished older people is the use of >25–30 g whey protein-enriched supplements [7], which may result in preserved or even increased muscle mass and strength [7,8]. We reported that in healthy older adults, when compared to younger adults, the acute suppression (up to 3 h following ingestion) of energy intake by protein administered orally or infused directly into the duodenum is less, resulting in an increase of overall energy and protein intake in the older adults [9,10,11]. In healthy, younger adults, protein is considered to be the most satiating macronutrient and protein-rich supplements and diets are often recommended as a weight loss strategy in obese, younger individuals. There is a lack of definitive evidence on their efficacy [12,13], especially in older adults.

In this study, we aimed to characterize the effect of ageing on the suppression of food intake at breakfast, lunch, and dinner over a time period of 9 h by a pre-breakfast whey protein load (30 g and 70 g) compared to a control drink in healthy younger and older men. We hypothesized that suppression of energy intake by whey protein when compared to control would be less in healthy older than younger adults, resulting in an increase in cumulative energy and protein intake in the older men.

## 2. Materials and Methods

### 2.1. Subjects

The study included 15 healthy younger men (mean ± standard error of the mean (SEM) age: 27 ± 1 years; body weight: 76.1 ± 2.0 kg; height: 1.73 ± 0.02 m; body mass index (BMI): 25.8 ± 0.7 kg/m^2^) and 15 healthy older men (75 ± 2 years; 80.7 ± 2.9 kg; 1.75 ± 0.01 m; 26.6 ± 0.8 kg/m^2^). Body weight and BMI of the younger and older men did not differ significantly (*P* > 0.05). Subjects were recruited by online advertisement and by flyers placed on notice boards at the University of Adelaide, Adelaide, Australia.

Exclusion criteria included smoking; alcohol intake of >2 standard drinks on >5 days per week; being vegetarian; intake of any illicit substance; use of prescribed or non-prescribed medications that may affect appetite, body weight, gastrointestinal function, or energy metabolism; food allergy(s); diabetes mellitus (fasting glucose concentration >6.9 mmol/L); epilepsy; gallbladder, pancreatic, cardiovascular, or respiratory diseases; significant gastrointestinal symptoms, disease, or surgery; any other illness deemed significant by the investigator; and an inability to comprehend the study protocol. Inclusion criteria included being weight stable (<5% fluctuation in their body weight) at study entry, as assessed by their self-reported weight in the preceding 3 months, and maintenance of usual physical activity level.

All subjects gave written informed consent for inclusion before they participated in the study. The study was conducted in accordance with the Declaration of Helsinki and the protocol was approved by the Ethics Committee of The Royal Adelaide Hospital (HREC/18/CALHN/132) and registered under trial registration number ACTRN12618000881235.

### 2.2. Protocol

Each participant was studied on three occasions, separated by ~3–10 days. On each occasion, they received, in a randomized order (using the method of randomly permuted blocks; www.randomization.com), a single drink of either flavoured water (control; ~2 kcal), 30 g whey protein (120 kcal), or 70 g whey protein (280 kcal). The drinks were equivolaemic (~450 mL) and contained different quantities of food-grade unflavoured whey protein isolate (Bulk Nutrients, Tasmania, Australia) dissolved in varying amounts of distilled water, sodium chloride, and low-calorie lime cordial (Bickford’s “diet lime” cordial) [11].

Volunteers arrived at the laboratory at ~8.00 a.m. after fasting for ~12 h overnight and refraining from strenuous exercise and alcohol for 24 h. The subjects were provided with a standard meal the night before each study day (beef lasagne, McCain Foods Pty Ltd., Wendouree, VIC, Australia ∼591 kcal). Subjects were told that we were assessing perceptions of appetite around the 3 meals, but not that we measured their food/energy intake.

At baseline (t = −5 min), perceptions of appetite were assessed by visual analogue scales (VAS) and the antral area of the stomach (cm^2^) was measured with a LogiqTM e-ultrasound machine (GE Healthcare Technologies, Sydney, NSW, Australia). Subsequently, the drink was administered at t = −2 min (~8.30 a.m.) and was served in an opaque cup to ensure that the volunteers were blinded. Participants were asked to ingest the drink within 2 min. Following consumption of the drink (t = 0 min), palatability of the drink and perceptions of appetite were assessed by VAS. The antral area of the stomach was measured at several time points between the drink and breakfast (t = 0, 5, 20, 35 min) and not thereafter. Energy intake was measured at breakfast (t = 35–65 min; ~9 a.m.), lunch (t = 275–305 min; ~1 p.m.), and dinner (t = 515–545 min; ~5 p.m.). Breakfast and lunch consisted of a cold buffet-style meal (Table 1) and dinner consisted of a warm meal and a small variation of buffet items (Table 2). Subjects were instructed to consume food until they were comfortably full. Before and after consumption of the meals, perceptions of appetite, in terms of hunger, fullness, desire to eat, and prospective food consumption, were assessed (t = 0, 5, 20, 35, 65, 80, 95, 275, 305, 320, 335, 515, 545, 560, 575 min). Subjects were not permitted to consume any food or drink between ingesting the study drink and the end of the study day, except at the breakfast, lunch, and dinner meals provided during the study day. Water intake in between meals was allowed, but not within 30 min before their next meal.

### 2.3. Measurements

The primary outcome of the study was ad libitum energy intake at the buffet-style meal and secondary outcomes include antral area and appetite.

#### 2.3.1. Energy Intake

To quantify the amount eaten, the weights of the food items were recorded before and after they was offered to the subjects [11]. Energy intake and macronutrient composition was calculated using commercially available software (Foodworks 3.01, Xyris Software, Highgate Hill, QLD, Australia). Absolute (kcal) and percentage suppression of energy intake (expressed as % of energy intake of the control day) by protein were calculated.

#### 2.3.2. Antral Area

Gastric emptying (gastric retention) was determined by measuring the antral area of the stomach. The circumference of the antral area was measured with a LogiqTM e-ultrasound machine (GE Healthcare Technologies, Sydney, NSW, Australia) by using a 3.5 C broad spectrum 2.5–4 MHz convex linear array transducer. Antral area (cm^2^) was determined with the use of a caliper and calculation program built into the ultrasound machine. Volunteers were seated on a chair and were asked to be still during the measurement. The transducer was positioned vertically to obtain a parasagittal image of the antrum, with the superior mesenteric vein and the abdominal aorta in a longitudinal section. If gastric contractions were observed, the acquisition was paused until the contraction wave had passed. To calculate meal retention in the whole stomach, the fasting antral area (measured at baseline) was subtracted from subsequent measurements performed after ingestion of the drinks [14]. Gastric retention was then calculated at a given time point as:Retention (%) = [AA(t) − AA(f)]/[AA(max) − AA(f)] × 100,
where AA(t) = antral area measured at a given time point, AA(f) = fasting antral area, and AA(max) = maximum antral area recorded after drink ingestion [11].

#### 2.3.3. Perceptions of Appetite and Palatability

Perceptions of appetite in terms of hunger, fullness, desire to eat, and prospective consumption were assessed by use of a VAS questionnaire [15]. The questionnaire consisted of 100 mm horizontal lines, where 0 represented that the sensation was “not felt at all” and 100 represented that the sensation was “felt the greatest.” Volunteers placed a vertical mark on each horizontal line to signify the strength of each sensation at the specified time points. Baseline fasting ratings were calculated as the mean of the three study days. Total AUC was calculated over 0–180 min [11].

Palatability of the drink was assessed by ratings of pleasantness, intenseness, full of taste, sweetness, saltiness, sour, bitterness, umami, and creaminess immediately after drink intake; palatability of the meal was assessed by like of taste, like of aftertaste, and enjoyability of the meal by use of a VAS questionnaire.

### 2.4. Data and Statistical Analysis

Statistical analyses were performed using SPSS software (version 24; IBM, Armonk, NY, USA). Power calculations were performed for the primary outcome of energy intake using measures of variance obtained from previous data (SD of 181 kcal) [11] to detect a minimum difference in suppression of energy intake by the treatment condition compared with the control of 251 kcal between younger and older subjects. Age and protein load main effects and the age by protein load interaction on outcomes were determined by using two-way repeated-measures analysis of variance (ANOVA). Residuals from all models were checked for normality and constant variance and all assumptions were found to be met. When significant treatment and/or interaction effects were present, Bonferroni corrected post hoc tests were performed to determine which specific drink conditions were different between age groups. Statistical significance was accepted at *P* < 0.05. All data are presented as means ± SEMs.

## 3. Results

The study protocol was well tolerated by all subjects.

### 3.1. Energy Intake

Energy intake after the drink (sum of breakfast, lunch, and dinner; Figure 1) was suppressed by whey protein compared to control (protein load main effect on energy intake *P* = 0.012), driven by the suppression of the 70 g whey protein drink (young: −251 ± 117 kcal, −8 ± 4%; older: −184 ± 96 kcal, −5 ± 4%; post-hoc test *P* = 0.023), which was greater (*P* = 0.027) when compared with the 30 g protein drink (young: −88 ± 108 kcal, −3 ± 4%; older: −5 ± 99 kcal, 0 ± 4%; Table 3). Suppression of energy intake by the 70 g whey protein compared to control (protein load main effect, *P* = 0.007) was greatest at lunch (young: −181 ± 83 kcal, −17 ± 8%; older: −154 ± 49 kcal, −15 ± 5%; *P* = 0.001; Figure 2). Protein intake of the drink, before breakfast, did not affect ad libitum energy intake at dinner in either age group. Suppression of energy intake (sum of breakfast, lunch, and dinner) by whey protein was less in healthy older men: −94 ± 82kcal when compared to younger men −169 ± 100 kcal (there was a main effect of age on suppression of energy intake by protein compared to control *P* = 0.027).

Cumulative energy intake (sum of energy in test drink, breakfast, lunch, and dinner) was not significantly different between study days and age groups (young: control: 2929 ± 131 kcal, 30 g whey protein: 2961 ± 161 kcal and 70 g whey protein: 2958 ± 163 kcal; older: 2878 ± 165 kcal, 2993 ± 122 kcal and 2974 ± 148 kcal, all *P* > 0.05).

### 3.2. Protein Intake

The sum of breakfast, lunch, and dinner protein intake after the test drinks decreased after the 70 g (*P* = 0.023), but not 30 g, whey protein drink when compared to the control day (protein load main effect *P* = 0.009, main effect of age *P* = 0.71, interaction effect *P* = 0.54).Cumulative protein intake (sum of protein in the drink plus protein intake at the meals) was increased in a protein load responsive fashion (young: control: 143 ± 10g, 30 g whey protein: +17%, 167 ± 9 g and 70 g whey protein: +36%, 195 ± 9 g; older: control: 133 ± 10 g, 30 g whey protein: +23%, 164 ± 10 g and 70 g whey protein: +47%, 195 ± 9 g; *P* < 0.001) comparably in the healthy younger and older men (main effect of age *P* = 0.71, interaction effect of age x protein load *P* = 0.54; Figure 3).

### 3.3. Gastric Emptying

Antral areas following overnight fasting (control: 3.4 ± 0.8 cm^2^; 30 g whey protein: 2.8 ± 0.7 cm^2^; 70 g whey protein: 2.9 ± 0.8 cm^2^; protein load main effect *P* = 0.21) and immediately after drink consumption (control: 15.6 ± 0.8 cm^2^; 30 g whey protein: 16.2 ± 0.8 cm^2^; 70 g whey protein: 16.4 ± 0.8 cm^2^; protein load main effect *P* = 0.76) were comparable between the study days for both the age groups. Gastric retention was greater after both protein drinks compared to control (main effect of age *P* = 0.27, protein load main effect *P* < 0.001, interaction effect *P* = 0.091; Figure 4).

### 3.4. Appetite

Baseline perceptions of appetite in terms of hunger (young: 61 ± 8 mm; older: 59 ± 9 mm), fullness (13 ± 4 mm; 5 ± 2 mm), desire to eat (61 ± 7 mm; 52 ± 8 mm), and prospective food consumption (67 ± 5 mm; 55 ± 6 mm) were not significantly different between study days and age groups after overnight fasting (all *P* > 0.05). Protein drink ingestion affected fullness (protein main effect *P <* 0.001), desire to eat (*P <* 0.001), and prospective food consumption (*P =* 0.002; Figure 5) in a protein load related fashion; fullness was higher (AUC, both *P <* 0.001) and desire to eat (AUC, *P =* 0.035 and *P =* 0.009) and prospective food consumption (immediately before lunch, *P =* 0.025, *P =* 0.006) were lower after the 70 g whey protein drink compared to control and the 30 g protein drink. Older compared to younger men had a lesser desire to eat (main effect of age *P =* 0.028) but also less fullness (main effect of age *P =* 0.003, interaction effect of age x protein load *P <* 0.001) throughout the day (Figure 5).

### 3.5. Palatability of Drinks and Meals

The 70 g whey protein drink was perceived to be creamier when compared to the flavored control drink (*P* = 0.016). Ratings of pleasantness, intenseness, full of taste, sweetness, saltiness, sour, bitterness, umami, and creaminess of the drinks were not significantly different (main effect of protein *P* > 0.05). The healthy younger men rated the drinks as more bitter than the older men (young: 19±4mm; older: 26 ± 3 mm, main effect of age *P =* 0.037). All other palatability ratings of the drinks were comparable between the age groups: pleasant (young: 47 ± 5 mm; older: 44 ± 4 mm), intense (51 ± 4 mm; 55 ± 3 mm), fullness (59 ± 4 mm; 59 ± 3 mm), sweet (53 ± 3 mm; 48 ± 3 mm), salty (31 ± 6 mm; 37 ± 4 mm), sour (34 ± 6 mm; 39 ± 4 mm), umami (34 ± 5 mm; 35 ± 3 mm), refreshing (40 ± 6 mm; 41 ± 4 mm), creaminess (27 ± 5 mm; 31 ± 3 mm, main effect of age all *P >* 0.05). Palatability of the meals, assessed as ratings of taste, aftertaste, and enjoyability, were comparable between study days and age groups (control, 30 g, 70 g protein: young: taste: 73 ± 5 mm, 75 ± 5 mm, 72 ± 6 mm, after taste: 73 ± 5 mm, 72 ± 5 mm, 73 ± 5 mm, enjoyable: 73 ± 5 mm, 75 ± 5 mm, 74 ± 5 mm; older: taste: 72 ± 4 mm, 74 ± 3 mm, 72 ± 4 mm, after taste: 71 ± 3 mm, 71 ± 3 mm, 72 ± 4 mm, enjoyable: 73 ± 4 mm, 76 ± 3 mm, 73 ± 4 mm; main effects of age, protein load main effects and interaction effects all *P >* 0.05).

## 4. Discussion

This study compared the acute effects of ingestion of whey protein drinks containing 30 g and 70 g to those of a flavored control drink consumed 35 min before breakfast on ad libitum energy intake at breakfast, lunch, and dinner, perceptions of appetite throughout the day, and gastric emptying (antral area) in healthy younger and older men. Energy intake (sum of breakfast, lunch, and dinner) was suppressed in a protein load-responsive fashion at breakfast and in particular, at lunch, but not at dinner. Suppression of combined energy intake at breakfast, lunch, and dinner by the protein drink was less in healthy older (−3%) when compared to younger (−7%) men. Cumulative protein intake (sum of protein drink plus protein intake at the meals) was increased in a protein load responsive fashion (+20% and +42%) in the healthy younger and older men. Gastric emptying of the protein drinks in the 35 min before breakfast was slower than that of the control. Fullness was higher and desire to eat and prospective food consumption lower after protein intake when compared with the control in a protein load related fashion. Older compared to younger men had a lower desire to eat but also lower fullness throughout the day, suggesting that older people experience lower sensitivity of the appetite-suppressing effects of a protein drink and may have a decreased perception of gastric distension as seen in our previous study [16,17].

Overall, suppression of energy intake by protein was less in healthy older than younger men in this study, confirming the results of our previous studies [11,18,19,20,21], e.g., in a study with a comparable design, suppression of energy intake by oral whey protein ingestion was ~−15% in healthy young compared to ~−1% in older men. In the present study, energy intake (sum of breakfast, lunch, and dinner) was suppressed most by the 70 g whey protein load compared to control (~7%) and at lunch, 4 h 35 min after the drink (~−20% in young and ~−15% in older men). In contrast, there was no suppression of energy intake by pre-breakfast protein at dinner time, 8 h 35 min after the drink, in either age group (~+7% compared to control dinner). We reported previously that in healthy older people, the timing of a 30 g whey protein drink (3 h, 2 h, 1 h, and immediately before the buffet-style meal) does not affect subsequent energy intake in older people. The effect of the whey protein ingestion on energy intake throughout the day may be associated with the slightly slower gastric emptying, reported by us and others in previous studies measuring gastric emptying for a period of 3 h in healthy older, when compared with younger, people [11,18,19,20]. Gastric emptying may be associated with postprandial satiety by affecting plasma gut hormone concentrations [22] in healthy younger adults [14,16,23,24].

The cumulative energy intake (sum of drink, breakfast, lunch, and dinner) was comparable between study days while cumulative protein intake was elevated during the protein conditions in both age groups. Cumulative energy intake on the protein days compared to control was slightly higher in older (+4%) than younger men (−1%), as was reported in our previous studies determining *ad libitum* energy intake 3 h after oral whey protein ingestion [11] and following 1 h whey protein infusions directly into the small intestine [9]. The insignificant effect of the whey drink on cumulative daily energy intake in this study may indicate that the ingestion of a single daily dose of whey protein, in doses up to 70 g, is unlikely to be a successful weight loss strategy to achieve a negative energy balance, without taking the effects on energy expenditure and muscle anabolism into account. Even if whey protein was given more than once a day, we have no evidence that this would have resulted in a greater cumulative energy deficit, particularly in older adults. The energy content of the protein drink would have equalled or outweighed suppression of energy intake produced by the protein drink. Given our finding with one protein drink before breakfast, it is likely that suppression of cumulative energy intake with multiple drinks would have been even less [25]. The subjects in this study were not aware, however, that we were interested in or measuring their ad libitum meal energy intake throughout the day in response to the different drinks. Young adults using protein supplements to lose weight may have different responses to those in this study. Cumulative protein intake was significantly increased by the 30 g and 70 g whey protein loads, particularly in the older men (young: +17% and +36% and older: +23% and +47%), reaching meaningful amounts sufficient to result in postprandial muscle anabolism in older adults [8,26]—the 70 g whey protein drink increased protein intake by 62 g, or ~0.8 g/kg body weight, in the older men.

A limitation of the study was that we only studied men. This was to enable comparisons with the results of our previous studies conducted in men which clearly showed the effect of protein load. As men generally show greater variations in appetite and food intake in response to energy manipulation than women [27,28], the effects of the protein drinks may be different in women and it would be appropriate to perform further studies including women. The healthy older participants were well nourished, unrestrained eaters, had an active lifestyle, and comparable energy intake on the control day to the younger men. It has been reported numerous times that healthy ageing is associated with reduced food intake [21,29] and hunger [21,30,31] and a blunting of the regulation of food intake [27,32] as suggested by the findings of this study, i.e., less suppression of energy intake by protein. The suppressive effect of whey protein in younger adults may be affected by having dietary restraints or actively trying to lose body weight [33,34]. Furthermore, the overall suppressive effect of protein supplements may be influenced by protein supplement intake before each meal of the day. The significant increase in cumulative protein intake and slight increase in cumulative energy intake in the older men suggests that whey protein can be given at breakfast, and possibly also at other meals, without decreasing overall daily energy intake, which would benefit malnourished, frail, older people—further studies are warranted. Another possible limitation was that the study was limited to 9 h after drink ingestion. As the effect of the pre breakfast drink on energy and protein intake had worn off by dinner, however, it seems unlikely that it would have had any effect after that.

## 5. Conclusions

Energy intake was suppressed by whey protein drinks in a protein load-responsive fashion at breakfast and particularly, at lunch, but not at dinner, and suppression of energy intake by protein was less in healthy older than younger men. Cumulative protein intake was increased in a protein load responsive fashion. These findings support the use of whey-protein drink supplements in healthy older patients who aim to increase their protein intake without decreasing their overall energy intake.

## Figures and Tables

**Figure 1 nutrients-12-03318-f001:**
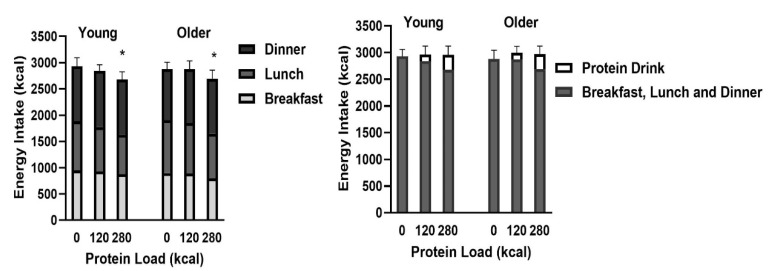
Energy intake at breakfast, lunch, and dinner following whey protein ingestion in healthy young and older men. Mean (± SEM) ad libitum energy intake (kcal; left) at breakfast (light grey bars), lunch (dark grey bars), and dinner (black bars) following drink ingestion containing flavored water (control, ~2 kcal) or whey protein (30 g/120 kcal or 70 g/280 kcal) and cumulative energy intake (kcal; right; sum total energy intake at breakfast, lunch, and dinner combined (dark grey bars) and protein drink (white bars)) in young (left; *n* = 15) and older (right; *n* = 15) men. Age and protein load main effects and interaction effects were determined by repeated measures ANOVA. * The 70 g protein drink suppressed energy intake (sum of breakfast, lunch, and dinner) compared with the control (protein load effect *P* = 0.012, post-hoc *P* = 0.023).

**Figure 2 nutrients-12-03318-f002:**
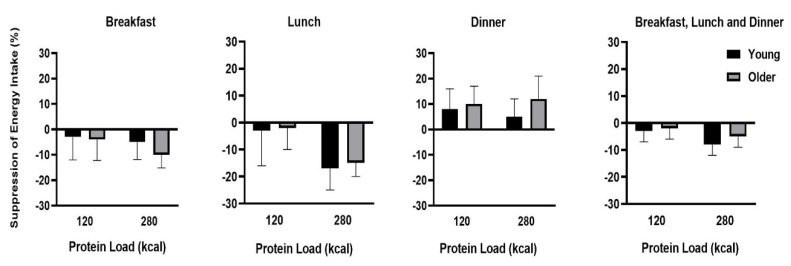
Suppression of energy intake by whey protein at breakfast, lunch, and dinner and total suppression of energy intake in healthy young and older men. Mean (± SEM) suppression of energy intake (kcal) at breakfast, lunch, and dinner following whey protein (30 g/120 kcal or 70 g/280 kcal) ingestion compared to control (~2 kcal) in young (black shading; *n* = 15) and older (grey shading; *n* = 15) men. Age and protein load main effects and interaction effects were determined by using repeated-measures ANOVA. Energy intake was suppressed by protein (protein load main effect *P* = 0.012). Suppression of energy intake by 70 g protein (*P* = 0.007) was evident, particularly at lunch (*P* = 0.001). Suppression of energy intake (sum of breakfast, lunch, and dinner) by protein was less in healthy older than younger men (main effect of age *P* = 0.027).

**Figure 3 nutrients-12-03318-f003:**
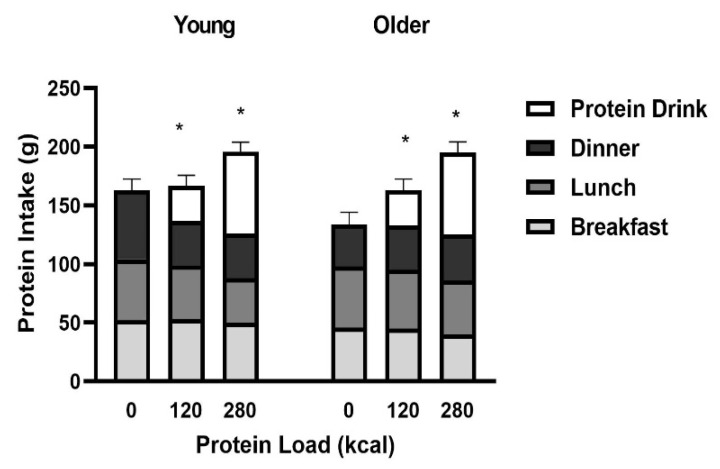
Mean (± SEM) protein intake (g) at breakfast (light grey bars), lunch (dark grey bars), and dinner (black bars) following drink ingestion containing flavored water (control, ~2kcal) or whey protein (30 g/120 kcal or 70 g/280 kcal; white bars) in young (left; *n* = 15) and older (right; *n* = 15) men. Age and protein load main effects and interaction effects were determined by using repeated-measures ANOVA. * Cumulative protein intake (sum of protein drink plus protein intake at meals) was increased in a protein load responsive fashion comparably in the healthy young and older men (main effect of age *P* = 0.71, protein load main effect *P* < 0.001, interaction effect *P* = 0.54).

**Figure 4 nutrients-12-03318-f004:**
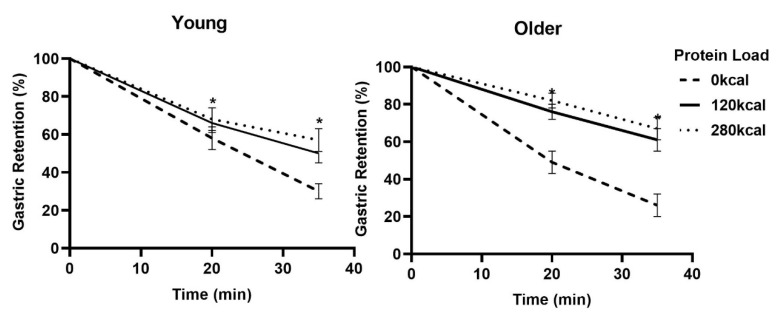
Mean (± SEM) Gastric Retention (%) of drinks containing flavored water (control, ~2 kcal) or whey protein (30 g/120 kcal or 70 g/280 kcal; open bars) in young (left; *n* = 15) and older (right; *n* = 15) men. Age and protein load main effects and interaction effects were determined by using repeated-measures ANOVA. * Gastric Retention, calculated based on the antral areas, were larger after both protein drinks compared to control (main effect of age *P* = 0.27, protein main effect *P* < 0.001, interaction effect *P* = 0.091).

**Figure 5 nutrients-12-03318-f005:**
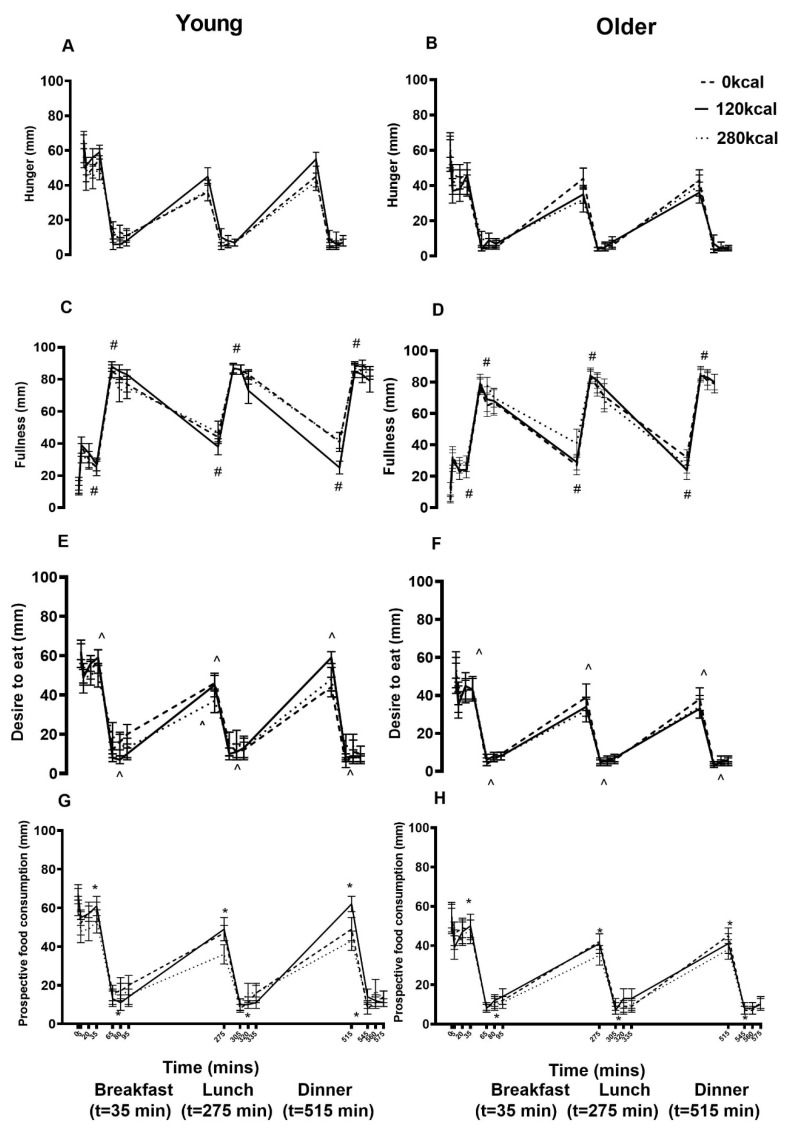
Mean (± SEM) visual analogue scores (VAS; 0–545 min) of hunger (**A**,**B**), fullness (**C**,**D**), desire to eat (**E**,**F**), and prospective food consumption (**G**,**H**) following overnight fasting (t = −5) and after drink ingestion (t = 0, 5, 20, 35, 65, 80, 95, 275, 305, 320, 335, 515, 545, 560, 575 min) containing flavored water (control, ~2 kcal) or whey protein (30 g/120 kcal or 70 g/280 kcal; open bars) and immediately before and after breakfast (B), lunch (L), and dinner (D) in young (left; *n* = 15) and older (right; *n* = 15) men. Age and protein load main effects and interaction effects were determined using repeated-measures ANOVA. Protein affected ^#^ fullness (protein load main effect *P* < 0.001), ^ desire to eat (*P* < 0.001), and * prospective food consumption (*P* = 0.002) in a protein load related fashion. Older compared to younger men had lower desire to eat (main effect of age *P* = 0.028) and fullness (*P* = 0.003, interaction effect *P* < 0.001).

**Table 1 nutrients-12-03318-t001:** Composition of the cold buffet-style breakfast and lunch meal.

Food Items	Amount Served (g)	Energy Content (kcal)	Protein (g)	Carbohydrate (g)	Fat (g)
Whole meal bread, 4 slices *	125	308	13.8	54.8	4.9
White bread, 4 slices *	125	304	11.1	61.4	2.7
Cheese, sliced ^†^	85	346	22.6	0.9	29.2
Ham, sliced ^‡^	100	95	17.1	3.5	1.8
Chicken, sliced ^§^	100	104	19.4	3.7	1.7
Margarine ^||^	20	108	0.0	0.0	12.4
Mayonnaise ^¶^	20	137	0.4	0.7	15.2
Tomato, sliced	100	13	1.0	2.0	0.1
Cucumber, sliced	100	11	0.5	2.0	0.1
Lettuce	100	5	0.9	0.4	0.0
Apple	170	89	0.5	2.0	0.1
Banana	190	166	3.3	39.0	0.2
Fruit salad **	140	81	0.4	17.7	1.3
Strawberry yogurt ^††^	175	162	9.1	25.0	3.4
Chocolate custard ^‡‡^	100	105	3.3	16.9	3.1
Milky Way ^§§^	12	52	0.3	9.0	1.9
Orange juice, unsweetened ^||||^	300	117	1.9	22.6	2.7
Iced coffee ^¶¶^	375	254	12.4	38.3	6.6
Water	600	0	0.0	0.0	0.0
Total		2457	19%	49%	32%

* Sunblest, Tiptop, George Weston Foods Ltd., Enfield, NSW, Australia. ^†^ Coon Tasty Cheese slices, Australian Cooperative Foods Ltd., Sydney Olympic Park, NSW, Australia. ^‡^ KR Castlemaine boneless leg ham, George Weston Foods Ltd., Enfield, NSW, Australia. ^§^ Inghams chicken breast, Inghams Enterprises Pty Ltd., Burton, SA, Australia. ^||^ Vita-Lite canola, Peerless Holdings Pty Ltd., Braybook, VIC, Australia. ^¶^ MasterFoods, Mars Food Australia, Berkeley Vale, NSW, Australia. ** Goulburn Valley, SPC, Ardmona Operations Ltd., Shepparton, VIC, Australia. ^††^ Yoplait, LD&D Foods Pty Ltd., Docklands, VIC, Australia. ^‡‡^ Yogo, LD&D Foods Pty Ltd., Docklands, VIC, Australia. ^§§^ Mars Chocolate Australia, Wendouree, VIC, Australia. ^||||^ Golden Circle Orange juice, Golden Circle Limited, QLD, Australia. ^¶¶^ Farmers Union, LD&D Foods Pty Ltd., Docklands, VIC, Australia.

**Table 2 nutrients-12-03318-t002:** Composition of the dinner meal.

Food Items	Amount Served (g)	Energy Content (kcal)	Protein (g)	Carbohydrate (g)	Fat (g)
Pasta with Meatballs ^¤^	500	720	27.7	78.4	35.0
Whole meal bread, 4 slices *	125	308	14.0	55.5	4.9
Margarine ^||^	20	108	0.0	0.0	12.5
Philadelphia cream cheese º	68	175	3.8	2.1	17.3
Apple	170	89	0.5	2.0	0.1
Banana	190	166	3.3	39.5	0.2
Fruit salad **	140	81	0.4	17.9	1.4
Strawberry yogurt ^††^	175	162	9.2	25.3	3.4
Chocolate custard ^‡‡^	100	105	3.3	17.1	3.1
Muesli bar ^Ͽ^	35	185	5.6	12.5	13.1
Orange juice, unsweetened ^||||^	300	117	1.9	22.9	2.7
Water	600	0	0.0	0.0	0.0
Total		2216	13%	49%	38%

^¤^ Man Size Spaghetti and Meatballs, McCain Foods Pty Ltd., Wendouree, VIC, Australia. * Sunblest, Tiptop, George Weston Foods Ltd., Enfield, NSW, Australia. ^||^ Vita-Lite canola, Peerless Holdings Pty Ltd., Braybook, VIC, Australia. º Philadelphia Spreadable Cream Cheese snack tubs, Consumer Advisory Service, Melbourne, VIC, Australia. ** Goulburn Valley, SPC, Ardmona Operations Ltd., Shepparton, VIC, Australia. ^††^ Yoplait, LD&D Foods Pty Ltd., Docklands, VIC, Australia. ^‡‡^ Yogo, LD&D Foods Pty Ltd., Docklands, VIC, Australia. ^Ͽ^ Coles Nut bars, choc coated, Coles Supermarkets Australia Pty Ltd., Hawthorn East, VIC, Australia. ^||||^ Golden Circle Orange juice, Golden Circle Limited, QLD, Australia.

**Table 3 nutrients-12-03318-t003:** Energy intake at and macronutrient composition of breakfast, lunch, and dinner following whey protein drink ingestion in healthy young and older men.

	Young (*n* = 15)				Older (*n* = 15)			
	Breakfast	Lunch	Dinner	Total	Breakfast	Lunch	Dinner	Total
Control drink								
Energy intake (kcal)	947± 64	933 ± 74	1049 ± 68	2929 ± 131	896 ± 74	1007 ± 62	975 ± 79	2878 ± 165
Fat (energy %)	34 ± 1	34 ± 2	36 ± 2		29 ± 2	33 ± 6	39 ± 2	
Carbohydrate (energy %)	43 ± 2	43 ± 2	50 ± 2		51 ± 2	46 ± 2	47 ± 2	
Protein (energy %)	23 ± 1	23 ± 1	14 ± 1		20 ± 1	21 ± 1	14 ± 1	
30 g (120kcal) protein drink								
Energy intake (kcal)	925± 67	848 ±89	1068 ± 48	2841 ± 161	888 ± 60	962 ± 84	1023 ± 66	2873 ± 122
Fat (energy %)	34 ± 2	30 ± 3	38 ± 2		30 ± 1	34 ± 1	38 ± 2	
Carbohydrate (energy %)	43 ± 2	48 ± 4	47 ± 1		51 ± 3	46 ± 2	48 ± 1	
Protein (energy %)	23 ± 1	22 ± 2	15 ± 0		19 ± 1	20 ± 1	14 ± 1	
70 g (280kcal) protein drink								
Energy intake (kcal)	874 ± 70	752 ± 85 *	1052 ± 56	2678 ± 163	794 ± 72	853 ± 69 *	1047 ± 82	2694 ± 148
Fat (energy %)	34 ± 1	27 ± 2	48 ± 2		30 ± 2	32 ± 1	38 ± 2	
Carbohydrate (energy %)	43 ± 2	54 ± 3	47 ± 1		51 ± 4	46 ± 2	48 ± 1	
Protein (energy %)	23 ± 1	19 ± 2	15 ± 0		19 ± 1	22 ± 1	14 ± 0	

Mean (±SEM) ad libitum energy intake (kcal) at and macronutrient composition (energy percentage) of breakfast, lunch, and dinner, following drink ingestion containing flavoured water (control, ~2 kcal) or whey protein (30 g/120 kcal or 70 g/280 kcal) in young (left; *n* = 15) and older (right; *n* = 15) men. Age and protein load main effects and interaction effects were determined by using repeated-measures ANOVA. * Energy intake was suppressed by protein compared to control (protein load main effect *P* = 0.012). Suppression of energy intake by 70 g protein compared to control (*P* = 0.007) occurred particularly during lunch (*P* = 0.001).

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
