# Peer review of "Whey Protein Drink Ingestion before Breakfast Suppressed Energy Intake at Breakfast and Lunch, but Not during Dinner, and Was Less Suppressed in Healthy Older than Younger Men"

_nutrients, 2020, doi:10.3390/nu12113318_

Round 1

Author Response

Reviewer 1

Aim of the present study was that of investigating the effects of a preload of whey proteins at two different doses (30g and 70g vs placebo) on ad-libitum energy intake (at breakfast, lunch and dinner) and cumulative protein intake in a cohort of young and old men. Shortly, energy intake was suppressed by whey proteins in a dose-dependent way at breakfast and particularly at lunch, but not at dinner; importantly, suppression of energy intake by whey proteins was less in old than young men. Cumulative protein intake was increased by whey proteins in a dose-dependent way, particularly in the old group.

The experimental design is robust and complex, denoting the Authors’ expertise in this field. The results are well presented, though not completely discussed.

Comments:

1) Did the Authors evaluate the gaussian distribution of each variable before using any parametric test? 

Line 177: “Residuals from all models were checked for normality and constant variance and all assumptions were found to be met.”

2) Did the Authors perform a power test before recruiting patients? This information should be reported in the manuscript.

Line 70: “Calculations have been performed for the primary outcome of energy intake using measures of variance obtained from previous data (SD of 181 kcal [Giezenaar C, Trahair LG, Rigda R, Hutchison AT, Feinle-Bisset C,Luscombe-Marsh ND, Hausken T, Jones KL, Horowitz M, Chapman I, et al. Lesser suppression of energy intake by orally ingested whey protein in healthy older men compared with young controls. Am J Physiol Regul Integr Comp Physiol 2015;309:R845–54.]) to detect a minimum difference in suppression of energy intake by the treatment condition compared with control of 251 kcal between younger and older subjects.”

3) The Authors state that whey proteins cannot represent a valid strategy against obesity. As in any test of the study a single dose was administered (i.e., only before breakfast), one might argue that, due to pharmacokinetic reasons, the anorexigenic effect of whey proteins (or the absorbed amino acids) was disappearing in the late hours (i.e., before dinner). As demonstrated in the Work by Rigamonti et al. (Nutrients. 2019 Jan 23; 11(2):247), which must be cited in the Discussion, repeated doses of whey proteins might be necessary (perhaps, only two doses in a day) to maintain an effective anorexigenic response. The Referee invites the Authors to discuss this issue.

We agree with the reviewer that we determined the acute effects following a single dose of whey protein. The study of Rigamonti et al. (Nutrients. 2019 Jan 23;11(2):247) was performed in a cohort of obese young women without a non-caloric control condition, which would allow to determine suppression of energy intake by the test drink compared to control, and determined energy intake 2.5 hours after drink ingestion. Line: 332-340The insignificant effect of the whey drink on cumulative daily energy intake in this study may indicate that the ingestion of a single daily dose of whey protein, in doses up to 70g, is unlikely to be a successful weight loss strategy targeting to achieve a negative energy balance, without taking the effects on energy expenditure and muscle anabolism into account. Even if whey protein was given more than once a day, we have no evidence that this would have resulted in a greater cumulative energy deficit, particularly in older adults. The energy content of the protein drink would have equaled or outweighed the suppression of energy intake produced by the protein drink. Given our finding with one protein drink before breakfast it is likely that suppression of cumulative energy intake with multiple drinks would have been even less.”

4) The Authors should try to explain the pathophysiological mechanisms underlying the differences between younger vs. older groups. For instance, the Reader might be interested in knowing the reason of the increased cumulative protein intake, a finding more evident in older men.

The increased cumulative protein intake results from the combined intake of protein in the drink plus protein intake at the meals. Since the suppression of energy intake was less in older than younger men, the increase in cumulative protein intake was more evident in older when compared to younger men.

5) Line 153: greatest”.

It is not entirely clear to us which the reviewer is referring to. The sentence on line 153 was: “Perceptions of appetite in terms of hunger, fullness, desire to eat, and prospective consumption were assessed by use of a VAS questionnaire.”

6) Line 156-158: a reference (e.g., a previous work by the Authors using a similar protocol) is necessary.

 Line: 169 Giezenaar, C.; Trahair, L.G.; Rigda, R.; Hutchison, A.T.; Feinle-Bisset, C.; Luscombe-Marsh, N.D.; et al. Lesser suppression of energy intake by orally ingested whey protein in healthy older men compared with young controls. Am J Physiol Regul Integr Comp Physiol. 2015, 309, R845-R54; Parker BA, Sturm K, MacIntosh CG, Feinle C, Horowitz M, Chapman IM. Relation between food intake and visual analogue scale ratings of appetite and other sensations in healthy older and young subjects. Eur J Clin Nutr 58: 212–218, 2004.

7) The old men had less fullness (satiety) than the young men (line 253). This finding seems to contradict the well-known anorexia of ageing (even in a physiological context).

The Authors should discuss this issue.

“Older compared to younger men had lower desire to eat but also lower fullness throughout the day, suggesting that older people experience lower sensitivity of the appetite-suppressing effects of protein drink and may have a decreased perception of gastric distension as seen in our previous study [Giezenaar, C.; Van der Burgh, Y.; Lange, K.; Hatzinikolas, S.; Hausken, T.; Jones, K.L.; et al. Effects of Substitution, and Adding of Carbohydrate and Fat to Whey-Protein on Energy Intake, Appetite, Gastric Emptying, Glucose, Insulin, Ghrelin, CCK and GLP-1 in Healthy Older Men-A Randomized Controlled Trial. Nutrients. 2018, 10, 113.; Hutchison, A.T.; Piscitelli, D.; Horowitz, M.; Jones, K.L.; Clifton, P.M.; Standfield, S.; et al. Acute load-dependent effects of oral whey protein on gastric emptying, gut hormone release, glycemia, appetite, and energy intake in healthy men. Am J Clin Nutr. 2015, 102, 1574-84.].” The lack of extend of increase in fullness in older, when compared to the younger, adults is likely to be associated with a reduced responsiveness to the suppressive effects of nutrients on appetite and energy intake as described by Rolls et al [Rolls BJ, Dimeo KA, Shide DJ. Age-related impairments in the regulation of food intake. Am J Clin Nutr 1995;62:923e931]. In this study, after overnight fasting, perceptions of appetite in terms of hunger, fullness, desire to eat, and prospective food consumption, were not significantly different between study days and age groups. Postprandially, i.e. after the drink/throughout the day, older compared to younger men had lesser desire to eat but also less fullness. We and others have reported previously that the regulation of energy intake may be diminished in the elderly; fullness, measured in a postprandial state, was less in older when compared with younger adults [Giezenaar C, Chapman I, Luscombe-Marsh N, et al. Ageing Is Associated with Decreases in Appetite and Energy Intake-A Meta-Analysis in Healthy Adults. Nutrients 2016;309:R845eR854.; Soenen, S.; Giezenaar, C.; Hutchison, A.T.; Horowitz, M.; Chapman, I.; Luscombe-Marsh, N.D. Effects of intraduodenal protein on appetite, energy intake, and antropyloroduodenal motility in healthy older compared with young men in a randomized trial. Am. J. Clin. Nutr. 2014, 100, 1108–1115. MacIntosh, C.G.; Sheehan, J.; Davani, N.; Morley, J.E.; Horowitz, M.; Chapman, I.M. Effects of aging on the opioid modulation of feeding in humans. J. Am. Geriatr. Soc. 2001, 49, 1518–1524.; Rolls, B.J.; Dimeo, K.A.; Shide, D.J. Age-related impairments in the regulation of food intake. Am. J. Clin. Nutr. 1995, 62, 923–931.; Schneider, S.M.; al-Jaouni, R.; Caruba, C.; Giudicelli, J.; Arab, K.; Suavet, F.; Ferrari, P.; Mothe-Satney, I.; van Obberghen, E.; Hébuterne, X. Effects of age, malnutrition and refeeding on the expression and secretion of ghrelin. Clin. Nutr. 2008, 27, 724–731.; VanWalleghen, E.L.; Orr, J.S.; Gentile, C.L.; Davy, B.M. Pre-meal water consumption reduces meal energy intake in older but not younger subjects. Obesity 2007, 15, 93–99.; Zhou, B.; Yamanaka-Okumura, H.; Adachi, C.; Kawakami, Y.; Inaba, H.; Mori, Y.; Katayama, T.; Takeda, E. Age-related variations of appetite sensations of fullness and satisfaction with different dietary energy densities in a large, free-living sample of Japanese adults. J. Acad. Nutr. Diet. 2013, 113, 1155–1164.]

8) Line 270-272: did the perception of creaminess significantly change? This is not clear.

Line: 170 Palatability of the drink including creaminess was assessed immediately after drink intake and creaminess was not rated significantly different between drinks or age groups.

9) Line 129: were calculated.

Corrected.

Reviewer 2 Report

This is a well-designed trial to examine the impact of early morning protein ingestion on energy intake in young versus older men.  There were three treatments in randomized order:  control (2 kcal), 30 g protein (120 kcal), and 70 g protein (280 kcal).  Participants were followed for 9 hours to record energy and protein intake at breakfast, lunch, and dinner (9 hours).  Note, any energy intake after 9 hours but before retiring was not captured. The data showed that there was no overall change in total energy intake during the subsequent 9 hours between treatments, although there were reductions in energy intake particularly at lunch reflecting compensation for the energy in the protein drinks.  These reductions in energy intakes were lower for the older vs. younger men but overall, energy intakes increased ~30 kcals (young men) or ~100 kcals (older men) on the days the protein drinks were consumed.  Protein intakes also increased on the days the protein drinks were consumed. 

Comments

The title should reflect the results of the study, that early morning protein loads suppressed breakfast and lunch energy intakes but not total daily energy intakes in healthy younger and older men. 

Please state that written consent was obtained (if this is the case – or state the consenting situation). 

The data are represented repeatedly in slightly different forms in the figures and tables (Fig 1-3, Table 1).  Please combine these data into a single table with means and SD provided.  The energy in the protein drinks need to be incorporated into the numbers since the drinks were not isocaloric. 

That only men were studied is stated as a limitation but this is not justified in methods.  Please justify why only men were tested in this study. 

It is a limitation that any energy intake after 9 hours but before retiring was not captured. This needs to be added. 

Author Response

Reviewer 2

This is a well-designed trial to examine the impact of early morning protein ingestion on energy intake in young versus older men.  There were three treatments in randomized order:  control (2 kcal), 30 g protein (120 kcal), and 70 g protein (280 kcal).  Participants were followed for 9 hours to record energy and protein intake at breakfast, lunch, and dinner (9 hours).  Note, any energy intake after 9 hours but before retiring was not captured. The data showed that there was no overall change in total energy intake during the subsequent 9 hours between treatments, although there were reductions in energy intake particularly at lunch reflecting compensation for the energy in the protein drinks.  These reductions in energy intakes were lower for the older vs. younger men but overall, energy intakes increased ~30 kcals (young men) or ~100 kcals (older men) on the days the protein drinks were consumed.  Protein intakes also increased on the days the protein drinks were consumed. 

Comments

The title should reflect the results of the study, that early morning protein loads suppressed breakfast and lunch energy intakes but not total daily energy intakes in healthy younger and older men.

Thank you for your suggestion. We changed the title accordingly: ‘Whey protein drink ingestion before breakfast suppressed energy intake at breakfast and lunch, but not during dinner, and was less in healthy older than younger men.”

Please state that written consent was obtained (if this is the case – or state the consenting situation.

Line 83: “All subjects gave written informed consent for inclusion before they participated in the study.

The data are represented repeatedly in slightly different forms in the figures and tables (Fig 1-3, Table 1).  Please combine these data into a single table with means and SD provided.  The energy in the protein drinks need to be incorporated into the numbers since the drinks were not isocaloric.

Table 1 represents the composition of the cold buffet-style breakfast and lunch meal, while figure 1 represents the results of the ad libitum energy intake at the meals, with the graph on the right depicting the energy intake including the caloric value of the drinks, as suggested by the reviewer. We believe that figure 2 is necessary as it presents the data of the individual meals as suppression of energy intake as a percentage of the control meal intake and figure 3 which presents the intakes of protein. In our opinion combining these results into one table would make it too laboursome for the reader and therefore we believe it is most clearly demonstrated by leaving the figures and tables.

 That only men were studied is stated as a limitation but this is not justified in methods.  Please justify why only men were tested in this study.

Line 66: “We studied only men in order to compare with our previous studies conducted in men which clearly showed the effect of protein load. Another possible reason is that men appear to have the greatest ability to regulate energy intake in response to energy manipulation ([14]); in younger women, the menstrual cycle may have a confounding effect on appetite and energy intake.”

It is a limitation that any energy intake after 9 hours but before retiring was not captured. This needs to be added.

Line 359: “Another possible limitation was that the study was limited to 9 hours after drink ingestion. As the effect of the pre breakfast drink on energy and protein intake had worn off by dinner, however, it seems unlikely that it would have had any effect after that.”

Round 2

Reviewer 2 Report

The edits made by the authors are satisfactory.  Good luck with the paper. 

Author Response

Please see the attachment (Rebuttal letter and modified manuscript with track changes)

The modified title is:

Whey protein drink ingestion before breakfast suppressed energy intake at breakfast and lunch, but not during dinner, and was less in healthy older than younger men
